# A High-Accuracy, High Anti-Noise, Unbiased Frequency Estimator Based on Three CZT Coefficients for Deep Space Exploration Mission

**DOI:** 10.3390/s22197364

**Published:** 2022-09-28

**Authors:** Weitao Lu, Lue Chen, Zhen Wang, Jianfeng Cao, Tianpeng Ren

**Affiliations:** 1Beijing Aerospace Control Center, Beijing 100094, China; 2Xinjiang Astronomical Observatory, Chinese Academy of Sciences, Urumqi 830011, China

**Keywords:** Doppler, unbiased frequency estimation, CZT, anti-noise ability, Mars exploration

## Abstract

Deep space exploration navigation requires high accuracy of the Doppler measurement, which is equivalent to a frequency estimation problem. Because of the fence effect and spectrum leakage, the frequency estimation performances, which is based on the FFT spectrum methods, are significantly affected by the signal frequency. In this paper, we propose a novel method that utilizes the mathematical relation of the three Chirp-Z Transform (CZT) coefficients around the peak spectral line. The realization, unbiased performance, and algorithm parameter setting rule of the proposed method are described and analyzed in detail. The Monte Carlo simulation results show that the proposed method has a better anti-noise and unbiased performance compared with some traditional estimator methods. Furthermore, the proposed method is utilized to process the raw data of MEX and Tianwen-1 satellites received by Chinese Deep Space Stations (CDSS). The results show that the Doppler estimation accuracy of MEX and Tianwen-1 are both about 3 millihertz (mHz) in 1-s integration, which is consistent with that of ESA/EVN/CDSN and a little better than that of the Chinese VLBI network (CVN). Generally, this proposed method can be effectively utilized to support Chinese future deep space navigation missions and radio science experiments.

## 1. Introduction

In deep space exploration, the high-accuracy frequency of the downlink carrier or tone signal is a very important link to Doppler measurement, which is an essential spacecraft tracking technique that could support orbit determination of spacecraft and also provide important data for planetary radio science experiments, such as the measurement of the gravity field and planetary Radio Occultation (RO) [1]. The required accuracy of frequency estimation is typically several mHz, of which measurement accuracy is much higher than in other applications such as communications and the global navigation satellite system (GNSS). The phase-locked loop (PLL) receiver is used to estimate the frequency and phase at the deep space station, of which performance is negatively affected by the low Signal-to-Noise Ratio (SNR) and high dynamics of spacecraft [2,3]. In such situations, whether the accurate Doppler frequency can be derived or not is a critical requirement for spacecraft orbit determination. In addition to PLL, the open-loop Doppler measurement technique is also a preferable choice. In this case, the carrier or tone signal of the deep space probe is processed by modern signal theory to extract high-accuracy Doppler observables. Currently, the existing open-loop algorithms have the disadvantages of complicated algorithms or large computation [4,5] or special hardware platforms [6,7]. Chen et al. (2021) presented a Doppler frequency retrieving method via local correlation and segmented model, which was successfully applied to Chinese deep space exploration [8]. At the same time, there is a little complexity in setting the segment parameters for various dynamics.

The Doppler measurement in the open-loop mode is equivalent to the frequency estimation of a complex exponential signal corrupted by white noise, which is a traditional problem in signal processing. It is generally known that a two-stage search is implemented in many classical algorithms to improve the estimation performance. Firstly, a coarse search with an N-point Fast Fourier Transform (FFT) is executed, and then an optimal search is conducted around the peak determined in the previous stage. Various frequency estimators have been described in the literature, which can be divided into two groups: iterative methods and non-iterative methods. Zakharov and Tozer (1999) [9] presented a simple algorithm that consists of an iterative binary search for the true signal frequency. However, it is necessary to assign the data value of zeroes to compensate for 1.5 times the length of the original data in order to approach the Cramer–Rao Low Bound (CRLB) [10]. Aboutanios and Mulgrew (2005) [11] presented two more efficient iterative frequency estimators by calculating ±0.5 Discrete Fourier Transform (DFT) coefficients and the asymptotic variance is only 1.0147 times the CRLB. In another way, the non-iterative methods are more compact and more effective. Rife et al. (1974) [10] presented a famous estimator by using the two biggest FFT samples and almost reaches the CRLB when the signal frequency coincides with a bin center. Quinn (1994) [12] presented a number of estimators that interpolate the true signal frequency using the two DFT coefficients on either side of the maximum bin. Macleod (1998) [13] also used the three biggest FFT samples for frequency search. Jacobsen et al. (2007) [14] suggested a simple relation for FFT domain frequency estimation. The suggestion is based on empirical observations and presented without a proof. Therefore, Candan (2011) [15] presented a derivation for Jacobsen’s formula and presented a bias correction, which was effective for high SNR values and could be used at any SNR level. In order to utilize more signal power, Orguner and Candan (2014) [16] presented a special estimator which used all the bins in the FFT spectrum for the frequency estimation. The estimator had an improved performance but needed a high SNR level. In addition to the FFT spectrum, the Chirp-Z Transform (CZT) is also commonly used for frequency estimation. Proakis (2021) [17] described the theory of CZT in detail. Furthermore, Granados-Lieberman (2009) [18], Chen (2010, 2021) [8,19], and Zhang (2019) [6] have already introduced the CZT to improve the frequency estimation performance. The current methods estimate the frequency, and they are still mostly dependent on searching the peak line position of the CZT spectrum, meanwhile, making the estimation accuracy directly constrained by the CZT spectrum resolution. More importantly, the real frequency of the signal is continuous, and because of the fence effect of CZT, the estimated frequency must be the integral multiples of the CZT resolution. Therefore, the methods we mentioned above, which are based on the peak line position searching of the CZT spectrum, are obviously biased estimators. Herein, we presented the theoretical expressions of frequency by deducing the mathematic relationship between the three largest CZT spectrum samples and constructed an unbiased frequency estimator.

In this paper, a high accuracy frequency estimator is proposed. The method can be conducted in two steps: coarse frequency estimation by FFT spectrum and optimal frequency estimation by utilizing the mathematic relationship of the three largest CZT spectrum samples. The Monte Carlo simulations are implemented, and the raw data of deep spacecraft are processed to verify the performance of the proposed method. The results show that it has a better performance at anti-noise ability, frequency estimation bias, and accuracy. This paper is structured as follows. In the following section, we describe the frequency estimation problem. In Section 3, the proposed method is presented in detail. Moreover, the statistics performance and parameter setting are analyzed. Furthermore, in Section 4, the performance of the proposed method is verified and compared with some traditional estimators by Monte Carlo simulations. In Section 5, the proposed method is applied to the Doppler measurement of deep spacecraft. Moreover, the error sources are discussed in the frequency estimation of Tianwen-1. Finally, the conclusions are presented in Section 6.

## 2. Problem Description

An ideal single complex exponential waveform can be modeled as follows:(1)$$x(n)=Ae^{j(2\pi \frac{nk_{0}}{Nf_{s}}+\phi_{0})},n=0,1,\ldots ,N-1,k_{0}=Nf_{0}/f_{s}$$
where *A* and *f*_0_ are the signal amplitude and frequency, respectively, and $\phi_{0}$ is the initial phase, *f*_s_ is the sampling frequency, and *N* is the sampling points number. Without the loss of generality, let *A* = 1. The FFT spectrum of *x*(*n*) can be calculated as following:(2)$$X(k)=\sum_{n=0}^{N_{\mathrm{FFT}}-1}x(n)e^{-j\frac{2\pi }{N_{\mathrm{FFT}}}kn}=\frac{\sin(\pi (k-k_{0}))}{\sin(\frac{\pi }{N_{\mathrm{FFT}}}(k-k_{0}))}e^{j[\phi_{0}-(1-\frac{1}{N_{\mathrm{FFT}}})(k-k_{0})\pi ]},k=0,1,\ldots ,N_{\mathrm{FFT}}-1$$
where *N*_FFT_ is the length of FFT, so the frequency spectrum resolution $\Delta f=f_{s}/N_{\mathrm{FFT}}$. Let *k*_p_ be the peak position of the *N*_FFT_-point FFT spectrum, the real frequency, *f*_0,_ can be expressed as $f_{0}=k_{0}\Delta f=(k_{p}+\delta_{\mathrm{FFT}})\Delta f$, where $\delta_{\mathrm{FFT}}\in (-0.5,0.5]$, meaning the frequency bias corresponding to the FFT spectrum bins. Once $\delta_{\mathrm{FFT}}$ is estimated, the final frequency is estimated as ${\hat{f}}_{0}=(k_{p}+{\hat{\delta }}_{\mathrm{FFT}})\Delta f$, where ${\hat{\delta }}_{\mathrm{FFT}}$ is the estimation of $\delta_{\mathrm{FFT}}$. So, our goal is to accurately estimate $\delta_{\mathrm{FFT}}$.

The classical estimators mentioned in Section 1 mainly use the peak FFT spectrum and its two neighbors because the three samples occupy most of the signal power [13]. Due to the fence effect and spectrum leakage of FFT, the estimation performance may be degraded when the signal frequency is located at the bin center or edge. On the other hand, the second and third peaks of the FFT samples are smaller than the maximum recurrent peak sample, which limits the estimator’s ability to anti-noise. Moreover, because the frequency estimator is nonlinear, there is an SNR threshold for keeping the estimation performance [20].

Different from that, the FFT must be sampled for the entire unit circle of the z-plane uniformly, and the Chirp-Z transform (CZT) may provide a more centralized capability to perform a local z-transform that starts from an arbitrary point and samples with arbitrary uniform spacing, which is very suitable to overcome the low SNR in deep space situation [6]. The spectrum comparison of CZT and FFT is shown in Figure 1. As we can see, the fence effect and spectrum leakage of CZT are comparatively reduced; meanwhile, the spectrum resolution is higher than that of FFT. Therefore, in this paper, we propose a novel method by using the first three CZT samples.

## 3. Materials and Methods

### 3.1. Proposed Estimator

The Z-transform of *x*(*n*) is
(3)$$X(i)=\sum_{n=0}^{N-1}x(n)z^{-n},i=0,1,\ldots ,M-1$$
where *M* is the Z-transform length. Let $z=AW^{-k}$, $A{=e}^{j\theta_{0}}$, $W=e^{j\varphi_{0}}$, we obtain the CZT of *x*(*n*),
(4)$$X(i)=\sum_{n=0}^{N-1}x(n)e^{-jn(\theta_{0}+i\varphi_{0})},i=0,1,\ldots ,M-1$$
where *M* becomes the spectrum zooming factor, and $\varphi_{0}=\frac{2\pi L}{NM}$, is the spectrum resolution of CZT, $\theta_{0}=2\pi \frac{k_{\mathrm{st}}}{N}$, is the start frequency of spectrum zooming, *k*_st_ is the FFT sample index; and *B* is the spectrum analysis bandwidth of CZT, *B* = 2π*L*/*N*, *L* is the spectrum bandwidth factor.

We can obtain the expression of CZT spectrum by algebraic derivation,
(5)$$X(i)=e^{j\left[\phi_{0}+\left((1-\frac{1}{N})(k_{0}-k_{\mathrm{st}}-LK/M\right)\right]}\frac{\sin\pi (k_{0}-k_{\mathrm{st}}-Li/M)}{\sin\frac{\pi }{N}(k_{0}-k_{\mathrm{st}}-Li/M)}$$

The magnitude of *X*(*i*) is
(6)$$\left|X(i)\right|=\left|\frac{\sin\pi (k_{0}-k_{\mathrm{st}}-Li/M)}{\sin\frac{\pi }{N}(k_{0}-k_{\mathrm{st}}-Li/M)}\right|=\left|\frac{\sin\frac{L\pi }{M}\left(i-\frac{M}{L}(k_{0}-k_{\mathrm{st}})\right)}{\sin\frac{L\pi }{MN}\left(i-\frac{M}{L}(k_{0}-k_{\mathrm{st}})\right)}\right|=\left|\frac{\sin\frac{L\pi }{M}\left(i-k_{0}^{\prime }\right)}{\sin\frac{L\pi }{MN}\left(i-k_{0}^{\prime }\right)}\right|$$

$k_{0}^{\prime }=\frac{M}{L}(k_{0}-k_{\mathrm{st}})$, which stands for the real index of the signal frequency in the CZT samples. If $i_{p}$ stands for the position of peak of CZT spectrum, and $k_{0}^{\prime }=i_{p}+\delta_{\mathrm{CZT}},\left|\delta_{\mathrm{CZT}}\right|\leq 0.5$, then the magnitudes of the three largest CZT spectrum lines are, respectively:(7)$$\left\{\begin{array}{l} \left|X(i_{p})\right|=\left|\frac{\sin\left(\frac{L\pi }{M}\delta_{\mathrm{CZT}}\right)}{\sin\left(\frac{L\pi }{MN}\delta_{\mathrm{CZT}}\right)}\right|=\frac{\sin\left(\frac{L\pi }{M}\delta_{\mathrm{CZT}}\right)}{\sin\left(\frac{L\pi }{MN}\delta_{\mathrm{CZT}}\right)}\\ \left|X(i_{p}+1)\right|=\left|\frac{\sin\left[\frac{L\pi }{M}\left(1-\delta_{\mathrm{CZT}}\right)\right]}{\sin\left[\frac{L\pi }{MN}\left(1-\delta_{\mathrm{CZT}}\right)\right]}\right|=\frac{\sin\left[\frac{L\pi }{M}\left(1-\delta_{\mathrm{CZT}}\right)\right]}{\sin\left[\frac{L\pi }{MN}\left(1-\delta_{\mathrm{CZT}}\right)\right]}\\ \left|X(i_{p}-1)\right|=\left|\frac{\sin\left[\frac{L\pi }{M}\left(1+\delta_{\mathrm{CZT}}\right)\right]}{\sin\left[\frac{L\pi }{MN}\left(1+\delta_{\mathrm{CZT}}\right)\right]}\right|=\frac{\sin\left[\frac{L\pi }{M}\left(1+\delta_{\mathrm{CZT}}\right)\right]}{\sin\left[\frac{L\pi }{MN}\left(1+\delta_{\mathrm{CZT}}\right)\right]} \end{array}\right.$$

It is easy to know that L≤M, and *N* is usually set to a larger integer number for an accurate coarse frequency estimation. In Section 3, we set *N* = 1024 for the Monte Carlo Simulation. Therefore, it is easy to hold that L≪MN. In view of $\left|\delta_{\mathrm{CZT}}\right|\leq 0.5$, we can achieve the following approximation, as shown in Formula (8).
(8)$$\sin\left(\frac{L\pi }{MN}\delta_{\mathrm{CZT}}\right)\approx \frac{L\pi }{MN}\delta_{\mathrm{CZT}},\sin\left[\frac{L\pi }{MN}\left(1-\delta_{\mathrm{CZT}}\right)\right]\approx \frac{L\pi }{MN}\left(1-\delta_{\mathrm{CZT}}\right),\sin\left[\frac{L\pi }{MN}\left(1+\delta_{\mathrm{CZT}}\right)\right]\approx \frac{L\pi }{MN}\left(1+\delta_{\mathrm{CZT}}\right)$$

Therefore, substituting the expression in Formula (8) into Formula (7), we can obtain
(9)$$\left\{\begin{array}{l} \left|X(i_{p})\right|\frac{L\pi }{MN}\delta_{\mathrm{CZT}}=\sin\left(\frac{L\delta_{\mathrm{CZT}}\pi }{M}\right)\\ \left|X(i_{p}+1)\right|\frac{L\pi }{MN}\left(1-\delta_{\mathrm{CZT}}\right)=\sin\left[\frac{L\pi }{M}\left(1-\delta_{\mathrm{CZT}}\right)\right]=\sin\frac{L\pi }{M}\cos\frac{L\delta_{\mathrm{CZT}}\pi }{M}-\cos\frac{L\pi }{M}\sin\frac{L\delta_{\mathrm{CZT}}\pi }{M}\\ \left|X(i_{p}-1)\right|\frac{L\pi }{MN}\left(1+\delta_{\mathrm{CZT}}\right)=\sin\left[\frac{L\pi }{M}\left(1+\delta_{\mathrm{CZT}}\right)\right]=\sin\frac{L\pi }{M}\cos\frac{L\delta_{\mathrm{CZT}}\pi }{M}+\cos\frac{L\pi }{M}\sin\frac{L\delta_{\mathrm{CZT}}\pi }{M} \end{array}\right.$$

We can achieve Equation (10) as follows
(10)$$\left|X(i_{p}-1)\right|\frac{L\pi }{MN}\left(1+\delta_{\mathrm{CZT}}\right)-\left|X(i_{p}+1)\right|\frac{L\pi }{MN}\left(1-\delta_{\mathrm{CZT}}\right)=2\cos\left(\frac{L\pi }{M}\right)\sin\left(\frac{L\delta_{\mathrm{CZT}}\pi }{M}\right)=2\cos\left(\frac{L\pi }{M}\right)\left|X(i_{p})\right|\left(\frac{L\pi }{MN}\delta_{\mathrm{CZT}}\right)$$

After the elimination of the public factors in Equation (10) and some mathematic deductions, we can find
(11)$$\left(2\cos\left(\frac{L\pi }{M}\right)\left|X(i_{p})\right|-\left(\left|X(i_{p}+1)\right|+\left|X(i_{p}-1)\right|\right)\right)\delta_{\mathrm{CZT}}=\left|X(i_{p}-1)\right|-\left|X(i_{p}+1)\right|$$

Finally, the frequency can be estimated by utilizing Equations (12)–(14).
(12)$${\hat{\delta }}_{\mathrm{CZT}}=\frac{\left|X(i_{p}-1)\right|-\left|X(i_{p}+1)\right|}{2\cos\left(\frac{L\pi }{M}\right)\left|X(i_{p})\right|-\left(\left|X(i_{p}+1)\right|+\left|X(i_{p}-1)\right|\right)}$$
(13)$${\hat{k}}_{0}=k_{\mathrm{st}}+\frac{L}{M}(i_{p}+{\hat{\delta }}_{\mathrm{CZT}})$$
(14)$${\hat{f}}_{0}=\frac{f_{s}}{N}\left(k_{\mathrm{st}}+\frac{L}{M}(i_{p}+{\hat{\delta }}_{\mathrm{CZT}})\right)$$

The proposed method herein can be conducted in the following three steps:Calculate the FFT of *x*(*n*) and make the coarse frequency estimation by searching the peak position;Set the CZT parameters, including *M*, *k*_st_, and *L*, and calculate the CZT of *x*(*n*);Search the peak position of CZT and make the optimal frequency estimation by using Formulas (12)–(14).

### 3.2. Analysis of Unbiased Performance

Suppose that the single complex exponential waveform in Formula (1) is contaminated by Gaussian white noise, which is expressed as follows:(15)$$y(n)=x(n)+w(n),n=0,1,\ldots ,N-1,k_{0}=Nf_{0}/f_{s}$$
where the noise terms *w*(*n*) is assumed to be zero mean, complex additive Gaussian white noise with a variance of *σ*^2^. Therefore, the SNR can be given as 1/*σ*^2^. The CZT of *y*(*n*) can be expressed as follows:(16)$$Y(i)=\sum_{n=0}^{N-1}\left[x(n)+w(n)\right]e^{-jn(\theta_{0}+i\varphi_{0})}=X(i)+W(i),i=0,1,\ldots ,M-1$$
where *W*(*i*) is the CZT of *w*(*n*). To acquire the unbiased performance of the proposed estimator, we firstly analyzed the statistical distribution of *W*(*i*), which can be expressed as Formula (17):(17)$$W(i)=\sum_{n=0}^{N-1}w(n)e^{-jn(\theta_{0}+i\varphi_{0})}=W_{R}(i)-jW_{I}(i)$$
where *W_R_*(*i*) and *W_I_*(*i*) are the real and imaginary part of *W*(*i*), respectively, denoted as:(18)$$\left\{\begin{array}{l} W_{R}(i)=\sum_{n=0}^{N-1}w(n)\cos\left[n(\theta_{0}+i\varphi_{0})\right]\\ W_{I}(i)=\sum_{n=0}^{N-1}w(n)\sin\left[n(\theta_{0}+i\varphi_{0})\right] \end{array}\right.$$

It is clear to see that *W_R_*(*i*) and *W_I_*(*i*) can be regarded as a linear combination of a series of Gaussian white noise sequences. Therefore, *W_R_*(*i*) and *W_I_* (*i*) are also the Gaussian distribution. The average value and variance of *W_R_*(*i*) are given by:(19)$$\left\{\begin{array}{l} E\left[W_{R}(i)\right]=E\left[w(n)\right]\sum_{n=0}^{N-1}\cos\left[n(\theta_{0}+i\varphi_{0})\right]=0\\ E\left[W_{I}(i)\right]=E\left[w(n)\right]\sum_{n=0}^{N-1}\sin\left[n(\theta_{0}+i\varphi_{0})\right]=0 \end{array}\right.$$
where *E* [•] means the expectation operator. Then we can deduce that,
(20)$$E\left[Y(i)\right]=E\left[X(i)\right]+E\left[W(i)\right]=E\left[X(i)\right]$$

In view of the Gaussian white noise and in the statistical sense, we can find
(21)$$E\left[\left|Y(l)\right|\right]=E\left[\left|X(l)\right|\right],\;l=i_{p}-1,i_{p},i_{p}+1$$

Based on Formulas (7) and (8), Formulas (15) and (16) can obtained,
(22)$$\left\{\begin{array}{l} E\left[\left|Y(i_{p}-1)\right|-\left|Y(i_{p}+1)\right|\right]=E\left[\left|X(i_{p}-1)\right|-\left|X(i_{p}+1)\right|\right]\\ E\left[\left|Y(i_{p}-1)\right|+\left|Y(i_{p}+1)\right|\right]=E\left[\left|X(i_{p}-1)\right|+\left|X(i_{p}+1)\right|\right]\\ 2\cos\left(\frac{L\pi }{M}\right)E\left[\left|Y(i_{p})\right|\right]=2\cos\left(\frac{L\pi }{M}\right)E\left[\left|X(i_{p})\right|\right] \end{array}\right.$$

Substituting Formula (22) into (12) and carrying out the necessary manipulations, we find that the mathematical expectation of frequency bias estimation equals its real value,
(23)$$E\left[{\hat{\delta }}_{\mathrm{CZT}}\right]=\delta_{\mathrm{CZT}}$$

Therefore, the proposed method of frequency estimation is unbiased under the Gaussian white noise condition.

### 3.3. Analysis of Parameter Setting

As mentioned in Section 2, *B* is the spectrum bandwidth of CZT, *B* = *L*Δ*f*, and *L* is the number of FFT spectrum intervals. In order to reduce the calculation capacity and improve the spectrum resolution of CZT, *L* should be set to the minimum of reasonable values to cover the real frequency. The distribution of the FFT spectrum lines with different frequency biases, *δ*_FFT_, is displayed in Figure 2. When −0.5 ≤ *δ*_FFT_ < 0, as shown in Figure 1a and Figure 2c, the real frequency is located between the (*k*_p_ − 1)th and the (*k*_p_)th spectrum line. When 0 < *δ*_FFT_ ≤ 0.5, as shown in Figure 2b,d, the real frequency is located between the (*k*_p_)th and the (*k*_p_ + 1)th spectrum line. Considering the random effect caused by noise and to improve the robustness of the algorithm, it is suitable to set *L* = 2, and the bandwidth of CZT is centered at *k*_p_.

## 4. Numerical Simulations and Comparison

The algorithms above were implemented and simulated. The number of samples used in the simulation is *N* = 1024. The sampling frequency *f*_s_ = 1024 Hz, so the spectrum resolution of FFT is 1 Hz. The base frequency of the signal was *f*_0_ = 120 Hz (the signal frequency changes from *f*_0_ with a certain step as follows). With the CZT spectrum factor *L* = 2, the start and end frequencies of CZT are 119 Hz and 121 Hz, respectively. The spectrum zooming factor of CZT, *M* = 10, means the CZT spectrum resolution is 0.2 Hz. When the frequency bias zone is 0–0.5 Hz, and the frequency changing step is 0.025 Hz, there are 21 frequency values in total. The frequency estimation error *σ**_f_* and bias Δ*_f_* are calculated as follows:(24)$$\left\{\begin{array}{l} \sigma_{f}=\frac{1}{N_{\delta }N_{\mathrm{MC}}}\sum_{i=1}^{N_{\delta }}\sum_{j=1}^{N_{\mathrm{MC}}}{\left({\hat{f}}_{i}-f_{i}\right)}^{2}\\ \Delta_{f}=\frac{1}{N_{\delta }N_{\mathrm{MC}}}\sum_{i=1}^{N_{\delta }}\sum_{j=1}^{N_{\mathrm{MC}}}\left({\hat{f}}_{i}-f_{i}\right) \end{array}\right.$$
where *N_δ_* is the number of frequency values, herein *N_δ_* = 21, and *N*_MC_ is the Monte Carlo simulation times, herein *N_MC_* = 10000. *f*_i_ = *f*_0_ + *δ_i_* means the true frequency value when the frequency bias changes, and *δ_i_* = (*i* − 1) × 0.025 Hz. The CRLB on each SNR condition can be calculated by using Formula (25) [10]
(25)$$f_{\mathrm{CRLB}}=\frac{\sqrt{6}f_{s}}{2\pi \left(N^{1.5}-N^{0.5}\right)SNR^{0.5}}$$
where *N* is the data length of the integration time and *SNR* is the Signal-to-Noise Ratio.

Figure 3 displays the simulation results on the frequency estimation bias and error under three *SNR* conditions. As depicted in Figure 3a, the frequency bias randomly distributes around 0 Hz when the frequency bias changes and the mean bias of the three *SNR* conditions are −0.1918 mHz, 0.0327 mHz, and −0.0621 mHz, respectively, which shows that the proposed estimator here is unbiased. From Figure 3b, we can find that the frequency estimation error almost reaches the CRLB, especially when the *SNR* values are comparatively high, such as *SNR* = −10 dB and 0 dB here. The ratios of the frequency estimation error to the CRLB are 1.0905, 1.0173, and 1.0095 when *SNR* = −18 dB, −10 dB, and 0 dB, respectively.

Furthermore, the frequency estimation performances were compared with traditional algorithms, including Quinn (1994) [12], Rife (1974) [10], MacLeod (1998) [13], Aboutanios and Mulgrew (2005) [11], and Candan (2011) [15], as well as the CRLB [10]. The simulation conditions were set as mentioned above. There are 21 frequencies in total for each *SNR* condition, and the final estimation bias and error are the mean value of all the 21 frequency conditions. Figure 4 shows the frequency estimation error and bias of the proposed and the five traditional estimators.

The results in Figure 4 can be analyzed from the following three aspects.

(1)Anti-noise ability. As can be noted from this figure, there is a visible threshold effect except for the proposed method. Taking MacLeod’s (1998) [13] method as an example, when SNR is higher than −13 dB, the estimation bias and error are significantly decreased, and the error is very close to CRLB. While the estimation bias and error of the proposed estimator are more stable, even when the SNR is lower than −13 dB, showing that the proposed estimator has a high anti-noise ability.(2)Estimation bias. When the SNR is larger than the threshold, which is about −13 dB in this simulation, the estimation bias of MacLeod (1998) [13] and Candan (2011) [15] is significantly decreased, and the mean biases are 0.1 mHz and 1 mHz, respectively. There are obvious biases for Quinn (1994) [12], Rife (1974) [10], and Aboutanios and Mulgrew (2005) [11] under the same simulation conditions. However, the estimation bias of the proposed method is about 1 mHz when SNR = −20 dB, and the mean estimation bias of all the simulation SNR conditions is about 0.06 mHz, which means that the bias performance of the proposed method is comparatively better.(3)Estimation error. Figure 4b shows that the frequency estimation errors of the five traditional algorithms tend to stable. Among them, the Macleod (1998) [13] algorithm has the best performance with a variance of about 1.1626 times of CRLB. The estimation errors of Candan (2011) [15] and Rife (1974) [10] are about 1.5352 and 2.8760 times of CRLB.But the variances of the proposed method are about 1.2323, 1.0168 and 1.0131 times of CRLB when SNR = −20 dB, −10 dB and 0 dB, respectively. The results in Figure 4b show that the proposed method is much closer to CRLB compared with other five methods.

From what has been discussed above, we may reasonably arrive at the conclusion that the proposed method has a better anti-noise ability, frequency estimation bias, and accuracy.

## 5. Results

In this section, the raw data obtained from the Mars Express and Tianwen-1 orbiter observation experiment were utilized to evaluate the frequency estimation performance when the two probes were both orbiting Mars. The observation experiments were simultaneously implemented by CDSN, which consists of three Chinese deep space stations, the Jiamusi (JM) station, the Kashi (KS) station, and the Argentina (AG) station [8].

Compared with the Monte Carlo simulation, the most significant difference in performing high-accuracy frequency estimation with digital raw data is the downlink signals of typical non-stationary due to the relative motion between the spacecraft and ground stations. The abovementioned frequency estimators, including the proposed method, are suitable for processing stationary signals with a constant frequency. Therefore, it is necessary to eliminate the Doppler Effect before frequency estimation [21].

### 5.1. The Elimination of Doppler Effect

Assume that the frequency of the deep spacecraft downlink signal satisfies the n-order polynomial model as follows,
(26)$$f(t)=a_{n}t^{n}+a_{n-1}t^{n-1}+\ldots +a_{1}t+a_{0}$$
where $\left\{a_{i}\right\},i=0,1,\ldots ,n$ are the frequency model coefficients. Considering phase is the time integral of frequency, we can achieve the corresponding phase model,
(27)$$\varphi (t)=2\pi \int f(t)dt+\varphi_{0}=2\pi \left(\frac{a_{n}}{n+1}t^{n}+\frac{a_{n-1}}{n}t^{n-1}+\ldots +\frac{a_{1}}{2}t^{2}+a_{0}t\right)+\varphi_{0}$$

Now the signal model can be constructed as Formula (28),
(28)$$x(t)=e^{j\left(2\pi \int f(t)dt+\varphi_{0}\right)}=e^{j\varphi (t)}$$

In order to obtain the model coefficients in Formula (26), we should process the measured raw data. Assume the observation time length is *T*, and the sampling interval is *T*_s_. The frequency of the raw data is coarsely estimated at the integration time *T*_p_, and the number of frequency estimation is *K* = [*T*/*T*_p_], where [*x*] denotes the nearest integer number of *x*. The estimation results are remarked as: (29)$$\hat{F}=\left\{{\hat{f}}_{k}\right\},k=1,2,\ldots ,K$$

The time scale can be constructed at *T*_p_ interval, $t_{\mathrm{scale}}=\left(i+0.5\right)T_{p},i=0,1,\ldots ,K-1$. Combine Formulas (26) and (29), and the frequency model can be obtained by using the Least Square Method,
(30)$$\left\{{\hat{a}}_{n},{\hat{a}}_{n-1},\ldots ,{\hat{a}}_{1},{\hat{a}}_{0}\right\}$$

Next, construct the time scale of the raw data at the sampling interval *T*_s,_ $t_{\mathrm{scale}}^{\prime }=\left(i+0.5\right)T_{s},i=0,1,\ldots ,N-1$, where *N* = [*T*/*T*_s_], denotes the total points of the sampled raw data. Let ${\hat{a}}_{0}=0$, and the phase model and signal model can be constructed as follows:(31)$$\varphi_{\mathrm{mdl}}(t)=2\pi \int f_{\mathrm{mdl}}(t)dt+\varphi_{0}=2\pi \left(\frac{{\hat{a}}_{n}}{n+1}t^{n}+\frac{{\hat{a}}_{n-1}}{n}t^{n-1}+\ldots +\frac{{\hat{a}}_{1}}{2}t^{2}\right)$$
(32)$$x_{\mathrm{mdl}}(t)=e^{-j\varphi_{\mathrm{mdl}}(t)}$$

Finally, we can find the residual data $x_{\mathrm{res}}(t)$:(33)$$x_{\mathrm{res}}(t)=x(t)x_{\mathrm{mdl}}(t)=e^{j\left(2\pi a_{0}t+\varphi_{0}+2\pi f_{\mathrm{bias}}(t)t\right)}$$
where *f*_bias_(*t*) is the frequency bias caused by the inaccuracy of the frequency model, which can be nearly eliminated by iterative processing, when this is performed, the residual data *x*_res_(*t*) become a nearly stationary signal and can be processed by the proposed method, then *a*_0_ in Formula (33) can be accurately estimated, which combines with the frequency model in Formula (30) to generate the frequency estimation of the spacecraft’s downlink signal.

### 5.2. Mars Express Experiment

Before the first Chinese Mars exploration mission of Tianwen-1 was carried out, the China National Space Administration (CNSA) cooperated with the European Space Agency (ESA) to verify and confirm the Mars probe navigation ability of CDSN in 2020. MEX, which was launched on 2 June 2003, and has been orbiting Mars since December 2003 [22], was utilized to test and verify the feasibility of orbit measurement and orbit determination at the distance between Earth and Mars by CDSS. The data used herein were provided with the MEX observation experiments undertaken on 28 June 2020, from the JM and KS stations. The data were sampled and recorded with 0.5 MHz bandwidth and 8-bit quantification, whose format was the Delta-DOR Raw Data Exchange Format (RDEF) [23]. The sky frequency of the downlink signal is in the X band (about 8.4 GHz). The FFT spectrum of MEX at the JM station is shown in Figure 5, in which the sole peak strands for the carrier of the downlink signal. The carrier signal is utilized to estimate the high accuracy of the Doppler frequency using the proposed method.

After the elimination of the Doppler effect, the estimation of the residual frequency is shown in Figure 6a, which is the estimation of *a*_0_ in Formula (33). In order to show the details of the stochastic characteristics, the estimation result minus its average value is shown in Figure 6b. We can see that the residual frequency results are mostly located in a region of ±10 mHz, indicating that the residual signal after Doppler effect elimination is comparatively stationary. The frequency estimation error in the presented situation is about 3.52 mHz.

The Doppler measurement results of the MEX obtained by the proposed method in this paper were compared with that of the MEX obtained by the Very Long Baseline Array (VLBA), European VLBI Network (EVN), and the Chinese VLBI network (CVN) as well as CDSN [3,6,8], respectively, as shown in Table 1. For quantitative comparison, the Doppler frequency results of MEX by the proposed method were obtained within 1-s integration. The average accuracy Doppler frequency is 3.14 mHz in 1-s integration.

The study by Rosenblatt et al. (2008) [3] showed that the Doppler accuracy of the MEX obtained by ESA and NASA was about 3.2 mHz in 1-s integration. These measurement results were obtained by the digital baseband receivers of ESA and NASA in the closed-loop mode. The study by Zhang et al. (2019) [6] showed that the average 7.0 mHz precision of MEX in 1-s integration was obtained by CVN. All of the above results of the MEX Doppler accuracy are displayed in Table 1; here, we can see that the accuracy of the proposed method in this paper is approximately consistent with EVN and VLBA, as well as with the previous work at CDSN, while is about two times better than CVN. Since the raw data processed in references [3,6,8] and in this paper were sampled and recorded by different ground station assemblies, the comparison results have proven to be feasible and effective for the frequency estimation of the proposed method.

### 5.3. Tianwen-1 Experiment

Tianwen-1 is the first Chinese deep probe to Mars in Martian science research, which was launched on 23 July 2020 [24]. The raw data were recorded when the CDSS supported the navigation mission of Tianwen-1. We selected the observation conducted by the JM and KS stations on 26 February 2021, to verify the proposed method further. At that time, Tianwen-1 was on an elliptical orbit with a periastron of ~280 km, apastron of ~57,815 km, and an orbital period of ~49.0 h. The sampling frequency was 100 kHz, the quantification bit number was 8, and the data format was RDEF. The spectrum of Tianwen-1 is shown in Figure 7, which was observed by the KS station. The carrier signal in the spectrum is apparent.

Figure 8 shows the residual frequency estimation of the Tianwen-1 raw data at the JM station on February 26. Table 2 displays the estimation results of both stations with different integration times, from which we can see that the Doppler estimation errors of the proposed method at the JM station are 2.97 mHz in 1-s integration, 1.86 mHz in 5-s integration, and 1.41 mHz in 10-s integration, respectively. Moreover, the Doppler estimation RMS at the KS station are 3.06 mHz in 1 s-integration, 1.85 mHz in 5 s-integration, and 1.55 mHz in 10 s-integration, respectively. The results are consistent with that of Chen et al. (2021) [8]. It is concluded that the frequency estimation results with an accuracy of about 3 mHz in 1-s integration can provide high accuracy orbit determination of the Mars probe and will be helpful for future Chinese deep space radio science experiments. 

Moreover, the Signal-to-Noise Ratios (SNR) of the received signal at both stations are also estimated, which are about 4.1 dB at JM and 2.3 dB at KS, respectively. The corresponding CRLB of frequency estimation was calculated and is displayed in Table 2. There are big gaps between the frequency estimation error and the CRLB. 

The CRLB reflects the lower band of an unbiased estimation under the white noise condition. The thermal noise of the antenna’s receiver is usually modeled as Gaussian white noise. Therefore, as the integration time progresses, the impact of thermal noise is reduced, and consequently, the frequency estimation error is lower. Considering the gap between estimation error and the CRLB, we deduce that the thermal noise is not the dominant error factor for the frequency estimation in Tianwen-1.

### 5.4. Error Sources Discussion

The main error sources of the Doppler estimation in deep space exploration include phase scintillation, thermal noise, and frequency stability of the oscillator at the station [25]. The total analyzed error of frequency estimation caused by the above three factors, *σ_f,_* can be calculated as follows:(34)$$\sigma_{f}=\sqrt{\sigma_{f,PS}^{2}+\sigma_{f,Wn}^{2}\sigma_{f,FS}^{2}}$$

The phase scintillation is acquired by the downlink carrier when passing through the solar corona and introduces a random error to the Doppler measurement, which can be approximated by the following equation,
(35)$$\sigma_{f,PS}=\left\{\begin{array}{l} {\left(\frac{0.53C_{\mathrm{band}}}{T_{p}{}^{0.35}{\left[\sin(\theta_{\mathrm{SEP}})\right]}^{2.45}}\right)}^{0.5},{\;0}^{\circ }<\theta_{\mathrm{SEP}}\leq 90^{\circ }\\ {\left(\frac{0.53C_{\mathrm{band}}}{T_{p}{}^{0.35}}\right)}^{0.5},{\;90}^{\circ }<\theta_{\mathrm{SEP}}\leq 180^{\circ } \end{array}\right.$$

*θ*_SEP_ is the Sun-Earth-Probe angle (SEP) and *T_p_* is the estimation integration time. The constant parameter *C*_band_ depends on the working mode and frequency band. If the spacecraft transmits a signal which is transmitted from the ground and the ground-based reference for the Doppler is the same one that drives the transmitter, the observation mode is ‘‘two-way’’. Three-way Doppler measurement is analogous to two-way mode, except that the downlink carrier is received at a different station than that from which the uplink carrier was transmitted. On the observation date of Tianwen-1, the KS station transmitted the uplink signal to Tianwen-1 in the X band and received the coherent frequency from Tianwen-1 in the X band, too. This means that the observation of the KS station utilized a two-way mode. At the same time, the JM station received the downlink signal in the X band by the three-way mode. In the two-way or three-way mode, the *C*_band_ takes values of 5.5 × 10^−6^ when both the uplink and downlink frequency are in the X band.

The SEP angle during the observation of Tianwen-1 is about 78°, as shown in Figure 9. The corresponding frequency error caused by the phase scintillation is depicted in Figure 10. We can see that the errors are 1.754 mHz, 1.324 mHz, and 1.172 mHz for the 1 s, 5 s, and 10 s integration time, respectively.

The second error factor is thermal noise on both the uplink and downlink for the two-way and three-way mode in the Phase-locked-loop situation. As mentioned before, the thermal noise is always modeled as Gaussian white noise, and the error performance of the proposed method is no more than 1.0102 times that of CRLB when SNR is higher than 0 dB. Therefore, the white noise performance of the Tianwen-1 observation can be evaluated by the CRLB and has also been displayed in Table 2.

The third error factor is the frequency stability of the oscillator at the station, which can be depicted as the Allan deviation of the carrier’s frequency source [26]. The Turnaround Light Time (TLT) for the Tianwen-1 observation is about 20 min. Moreover, the traditional Allan deviation of the hydrogen maser at 1000 s is about 2 × 10^−15^. The Doppler measurement error caused by frequency source stable, *σ_f,FS_*, can be calculated as the following formula,
(36)$$\sigma_{f,FS}=\sqrt{2+\log_{2}\left(M\right)}f_{\mathrm{sky}}\sigma_{A}(\tau )$$
where *f*_sky_ is the sky frequency of downlink carrier, *M* equals to TLT/*τ*, and *σ*_A_(*τ*) is the Allan deviation at time interval of *τ*. Therefore, the frequency estimation error caused by frequency source stability in the Tianwen-1 observation is about 0.03 mHz.

Based on the above analysis, we can achieve the total analyzed error by using the Formula (34). For simplicity, only the results at the JM station are shown in Table 3. As we can see, the total analyzed errors are obviously smaller than the estimation errors. The ratios of the total analyzed error to estimation error are about 65%, 71%, and 83%, respectively. That is, the longer the integration time is, the closer the total analyzed error is to the estimation error. Since the thermal noise is related to the integration time, it is reasonable to deduce that the thermal noise performance of the proposed method is not comparable to but worse than CRLB when processing the raw data from the deep spacecraft. On the one hand, the thermal noise during the observation is possibly not ideal Gaussian white noise. On the other hand, the residual frequency of the raw data after the Doppler Effect elimination is still probably randomly changing, as depicted in Figure 6 and Figure 8, which means that the residual signal is not ideally stationary. Therefore, CRLB could not perfectly reflect the thermal noise effect in this situation.

In addition to the above three error factors, the terrestrial troposphere and ionosphere may also induce frequency estimation errors because of their temporal and spatial variety. It is known to all that tropospheric and ionospheric delay change with the observing elevation angle, changes with the motion of spacecraft. This means the tropospheric and ionospheric delay changes along with the motion of spacecraft during the observation. Figure 11 shows the tropospheric and ionospheric delay during the observation at the JM station, which is measured using the water vapor radiometer and the GNSS receiver with a sampling time of 1 s. It is easy to find that the tropospheric delay is time-varying, and the difference of tropospheric delay changes obviously with a random error of about 11.8 ps/s, which corresponds to a frequency error of about 95.6 mHz for the X band signal, much larger than that of frequency estimation error in Table 2. This is because, apart from the real variety of tropospheric delay, the water vapor radiometer has its own measuring accuracy and probably covers up the real variety of tropospheric delay. There is a similar phenomenon for the ionospheric delay but with a random error of about 7.2 mHz. Even so, it is reasonable to say that the tropospheric and ionospheric delay are also the error sources of the frequency estimation of the deep spacecraft downlink signal.

## 6. Conclusions

This paper presents a novel frequency estimator for Doppler measurement in deep space exploration. The proposed method is carried out with an FFT-based coarse frequency estimation and a fine estimation by utilizing the mathematic relation of the three CZT coefficients around the peak lobe. Firstly, the theoretical algorithm and signal processing procedures are described in detail. Monte Carlo simulations were implemented, and the results show that the unbiased frequency estimation error closely follows the CRLB in a lower SNR region in comparison to the previous estimators, including Rife (1974) [10], Macleod (1998) [13], Aboutanios and Mulgrew (2005) [11], and Candan (2011) [15], which indicate that the proposed frequency estimator has a better performance at anti-noise ability, frequency estimation bias, and accuracy. Then, the proposed method was utilized to process the received raw data of MEX and Tianwen-1 at the CDSS. The results show that the frequency estimation error of MEX and Tianwen-1 are both about 3 mHz in 1 s integration time. The accuracy of the Doppler frequency retrieving of MEX is consistent with ESA/EVN and is about two times better than CVN. Additionally, we evaluate the main error sources, including phase scintillation, frequency stability, and thermal noise, finding that phase scintillation is the dominant error source. However, there are some uncertain factors to be analyzed, such as the effects of tropospheric and ionospheric delay. Generally speaking, the proposed method herein can be effectively utilized to apply to future Chinese deep space navigation missions and can be a powerful support for radio science experiments in deep space exploration.

## Figures and Tables

**Figure 1 sensors-22-07364-f001:**
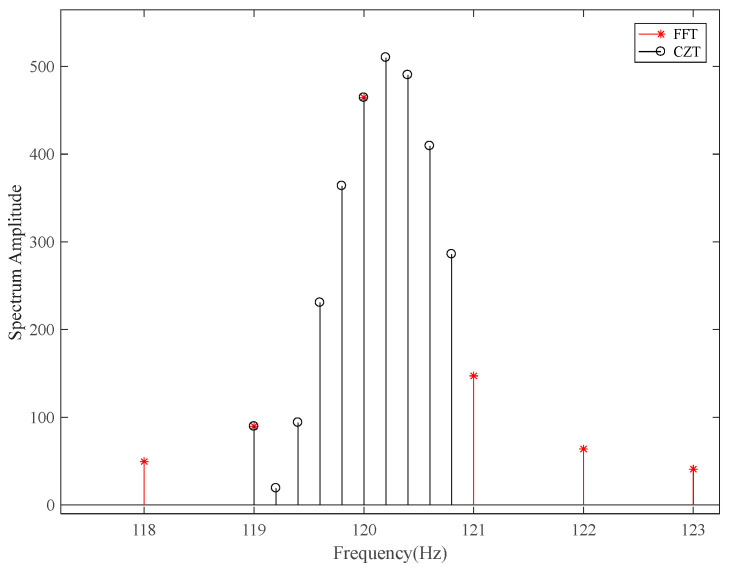
The spectrum comparison of CZT and FFT.

**Figure 2 sensors-22-07364-f002:**
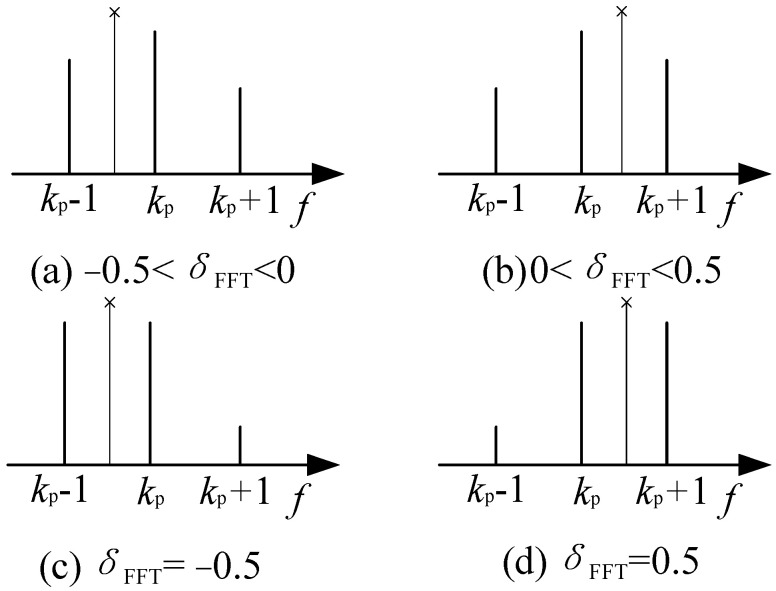
FFT spectrum lines with various δ_FFT_ (real frequency is marked by an oblique cross).

**Figure 3 sensors-22-07364-f003:**
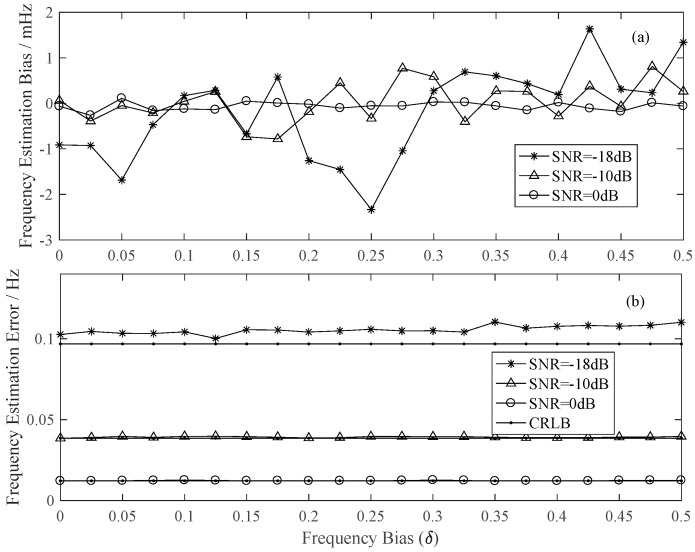
The estimation performance of proposed method under different SNR situations, (**a**) Frequency estimation bias, (**b**) Frequency estimation error.

**Figure 4 sensors-22-07364-f004:**
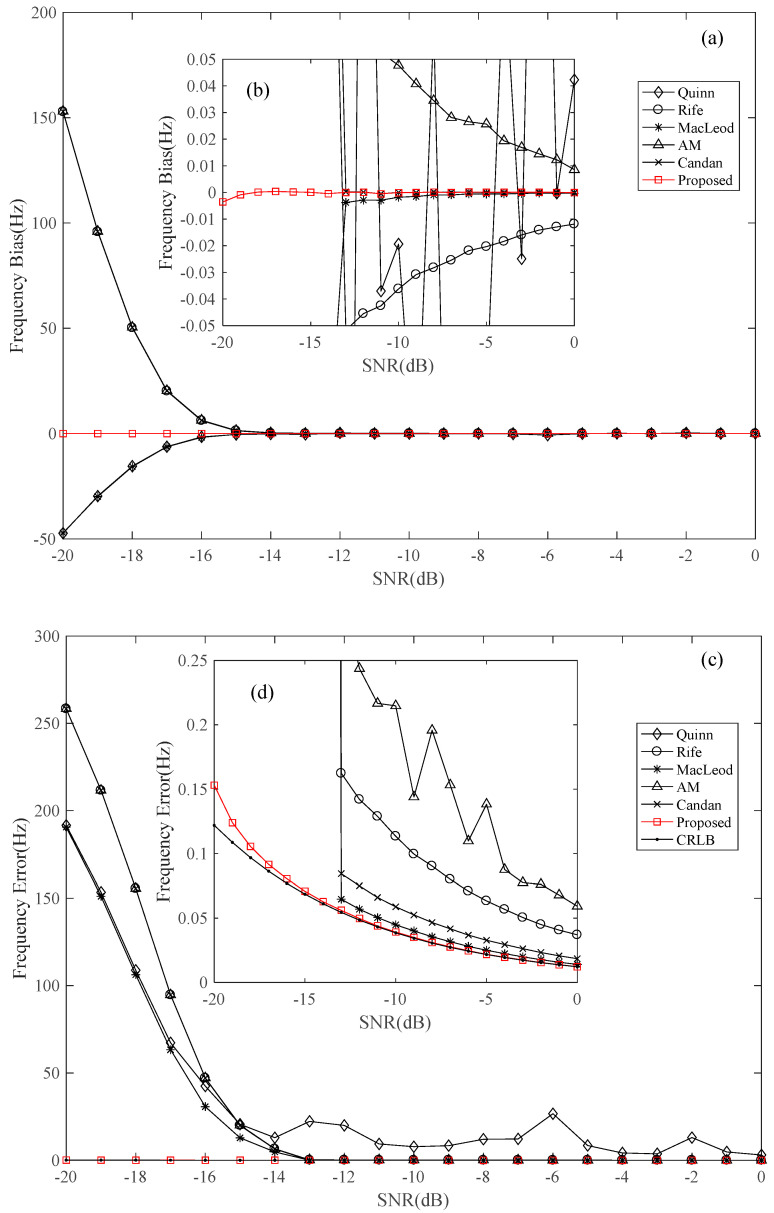
Frequency estimation performance compared with five traditional estimators and CRLB. (**a**) Frequency estimation bias; (**b**) the enlarged view of (**a**); (**c**) Frequency estimation error; (**d**) the enlarged view of (**c**).

**Figure 5 sensors-22-07364-f005:**
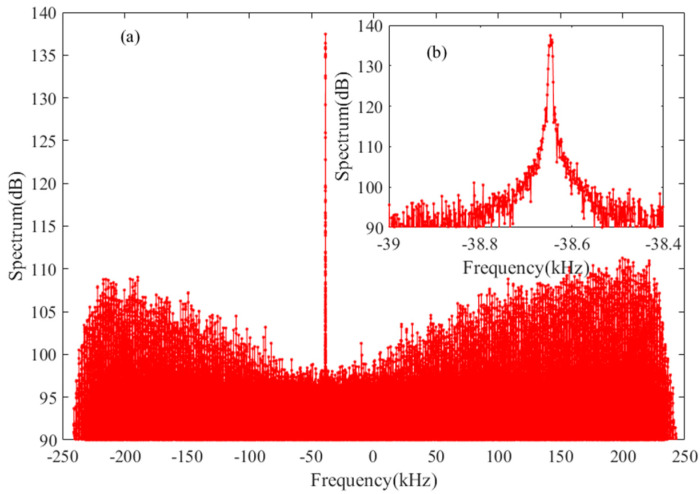
Spectrum of MEX observed at the JM station on 28 June 2020, with sampling frequency of 500 kHz (**a**) the whole spectrum (**b**) the enlarged view of carrier zone.

**Figure 6 sensors-22-07364-f006:**
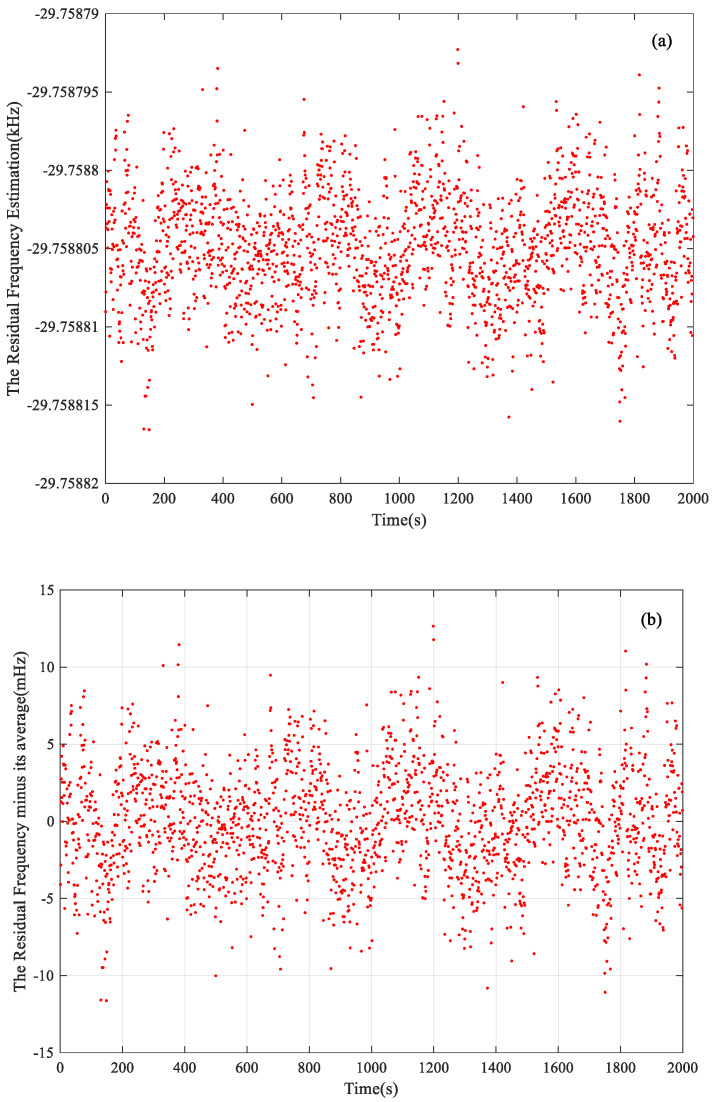
The residual frequency estimation of MEX raw data at JM station, (**a**) the residual frequency estimation; (**b**) the estimation result minus its average value.

**Figure 7 sensors-22-07364-f007:**
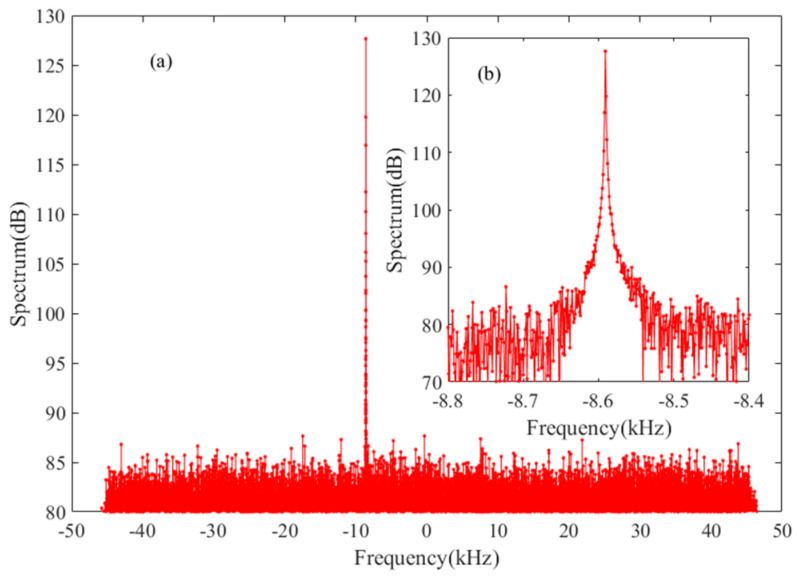
The spectrum of Tianwen-1 observed at the KS station on 26 February 2021, with sampling frequency of 100 kHz. (**a**) the whole spectrum; (**b**) the enlarged view of carrier zone.

**Figure 8 sensors-22-07364-f008:**
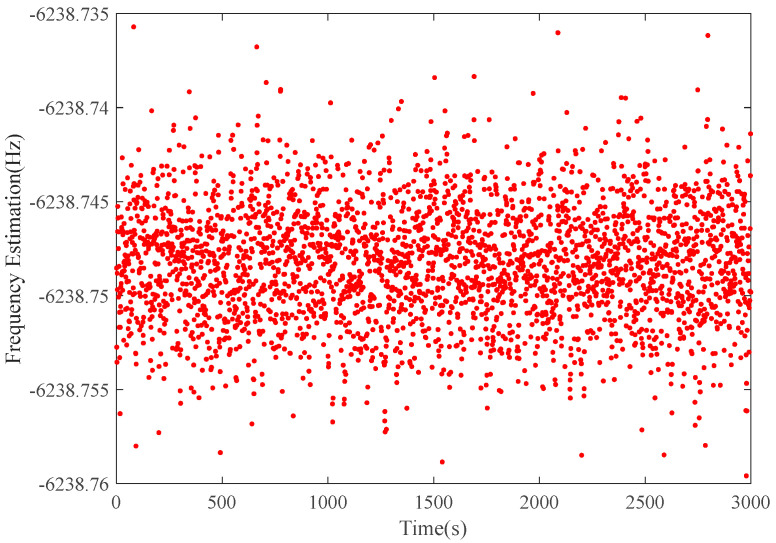
The residual frequency estimation of Tianwen-1 raw data at JM station.

**Figure 9 sensors-22-07364-f009:**
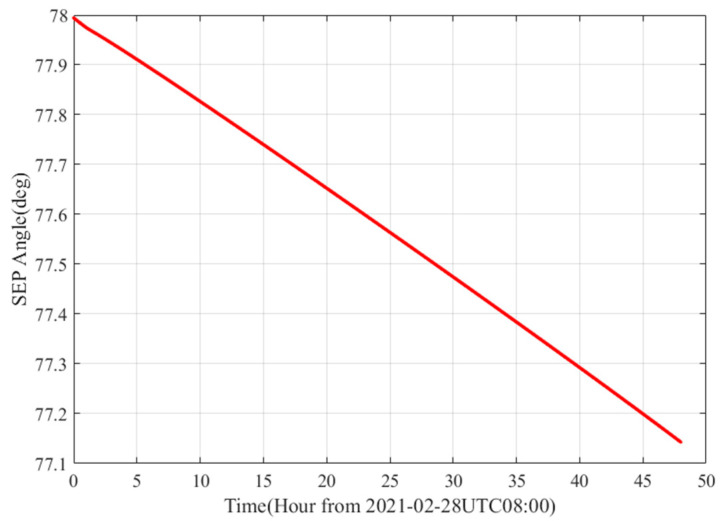
The SEP (Sun–Earth–Probe) of Tianwen-1 during observation period.

**Figure 10 sensors-22-07364-f010:**
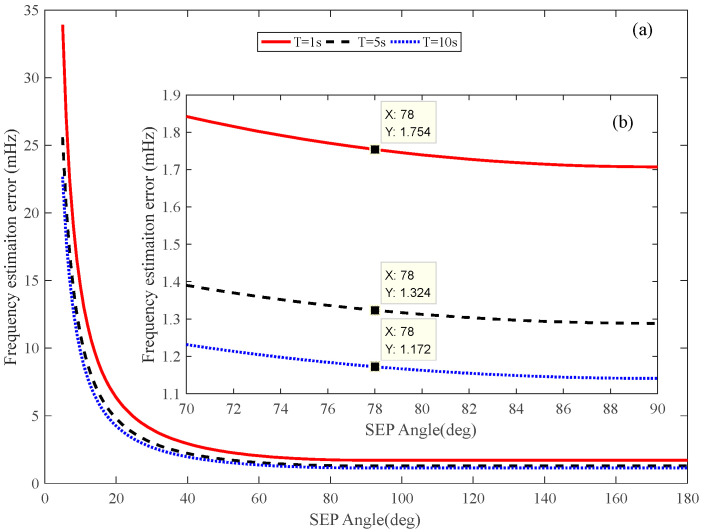
The frequency estimation error caused by phase scintillation (two-way and tree-way mode, X band). (**a**) full range of SEP angle; (**b**) enlarged view of SEP angle.

**Figure 11 sensors-22-07364-f011:**
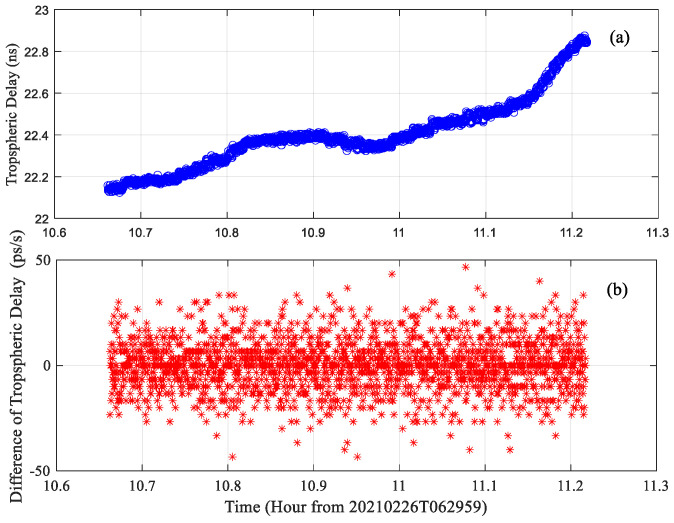
The tropospheric and ionospheric delay at JM station during the observation (**a**) tropospheric delay; (**b**) difference of tropospheric delay; (**c**) ionospheric delay; (**d**) difference of ionospheric delay.

**Table 1 sensors-22-07364-t001:** Frequency error of MEX (mHz, 1 s integration).

Station	EVN/VLBA	CVN	CDSN	CDSN (This Work)		
Accuracy	3.2	7.0	3.3	3.52	2.86	3.14
Remark	Ref. [3]	Ref. [6]	Ref. [8]	JM	KS	Average

**Table 2 sensors-22-07364-t002:** The frequency estimation results of Tianwen-1 (observed on 26 February 2021).

ID	Station	Estimated SNR (dB)	Integration Time (s)	Estimation Error (mHz)	CRLB (mHz)
1	JM	4.1	1	2.97	0.77
2			5	1.86	0.07
3			10	1.41	0.02
4	KS	2.3	1	3.06	0.95
5			5	1.85	0.08
6			10	1.55	0.03

**Table 3 sensors-22-07364-t003:** The comparison of estimation error and total analyzed error at JM station.

ID	Integration Time	Phase Scintillation	Thermal Noise	Frequency Source Stable	Total Analyzed Error	Estimation Error
1	1 s	1.754 mHz	0.77 mHz	0.03 mHz	1.916 mHz	2.97 mHz
2	5 s	1.324 mHz	0.07 mHz		1.326 mHz	1.86 mHz
3	10 s	1.172 mHz	0.02 mHz		1.173 mHz	1.41 mHz

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
