# Peer review of "A High-Accuracy, High Anti-Noise, Unbiased Frequency Estimator Based on Three CZT Coefficients for Deep Space Exploration Mission"

_sensors, 2022, doi:10.3390/s22197364_

Round 1
Reviewer 1 Report
I have no relevant comments, but it is not very clear
Why you selected Monte Carlo to carry out the simulations?
Author Response
Dear reviewer,
Thanks very much for your professional and detailed comments of sensors-1903723 entitled “A High-accuracy, High anti-noise, Unbiased Frequency Estimator Based on Three CZT Coefficients for Deep Space Exploration Mission”. These comments help us to improve the manuscript. We revised our manuscripts following the opinions point by point. Changes to the manuscript are highlighted in red bold font. Thanks again for your kind help.
The Doppler measurement in open-loop mode is equivalent to the frequency estimation of a complex exponential signal, which is a classical problem in signal processing, and there are many traditional methods, from which we find that the Monte Carlo simulations are widely used to verify the estimation performance, such as the work of Aboutanios and Mulgrew (2005), Orguner (2014), and so on.
By using Monte Carlo simulations, we can set an ideal signal with firm parameters, such as the frequency, and then add Gaussian random white noise to simulate the real signal in the deep space exploration. Finally, using the parameter estimation algorithm, we can evaluate the estimation performance through multiple iterations, such as the estimation accuracy, bias, and anti-noise ability.
Therefore, we adopt the Monte Carlo simulations to verify and compare the performance with several traditional methods by setting the same simulation parameters. On the other hand, we utilize the observational data transmitted by MEX and Tianwen-1 spacecraft to conduct the Doppler measurement, and compare the estimation results with previous work in the literatures to verify the performance of the proposed method more comprehensively.

Reviewer 2 Report
The paper is to introduce a novel method for frequency estimation for the Doppler measurement. Author used it for the Doppler measurement in Chinese Mars mission. The method is carried out in two steps: 1) a coarse FFT-based frequency estimation and 2) set Chirp-Z transform parameters and a fine estimation by usage of a relation of three Chirp-Z transform coefficient around the peak. The authors in the article showed the approbation of the method using Monte Carlo simulation of 1024 samples and compared the result with traditional algorithms described in the provided references. The authors claim that their proposed method has better performances at anti-noise ability, frequency estimation bias and accuracy.
The proposed method was successfully applied for processing the raw data from MEX and Tianwen-1 missions received by Chinese Deep Space Station. The frequency estimation errors were 3 mHz at 1 s measurement time. The author claim it is two time better than that form Chinese VLBI network. As the main source of the errors, the phase scintillations were defined.
An error message I mentioned on page 3: Bookmark is not defined.
I recommend this article for publication after a minor revision.

Author Response
Dear reviewer,
Thanks very much for your professional and detailed comments of sensors-1903723 entitled “A High-accuracy, High anti-noise, Unbiased Frequency Estimator Based on Three CZT Coefficients for Deep Space Exploration Mission”. These comments help us to improve the manuscript. We revised our manuscripts following the opinions point by point. Changes to the manuscript are highlighted in red bold font. Thanks again for your kind help.
Thanks very much for your approval of our paper and we highly appreciate you for reminding of the wrong bookmark in page 3. During the manuscript preparation, we used the cross-reference function of Microsoft Office Word when referring the literatures. Before submission, we modified the format according to the template and changed the citation index number to corresponding number. It is during the process when we omit the bookmark in page 3. Now we have already corrected the error.

Reviewer 3 Report
The submitted paper is very well written.
I have only one remark: The use of chirp-z transform in improving frequency estimation is even reported in classical signal processing books such as the Proaky's one. I consider that your paper is very complete because of the consideration of termal phenomena. Could you please add a disclaimer on your contribution telling chirp-z transform has been already used in improving the frequency but remarking your differences with respect to previous works, such as your remarkable results.
Please note that There is a bold text in page 3 signaling an error between square brackets.
Author Response
Dear reviewer,
Thanks very much for your professional and detailed comments of sensors-1903723 entitled “A High-accuracy, High anti-noise, Unbiased Frequency Estimator Based on Three CZT Coefficients for Deep Space Exploration Mission”. These comments help us to improve the manuscript. We revised our manuscripts following the opinions point by point. Changes to the manuscript are highlighted in red bold font. Thanks again for your kind help.
Question 1
The submitted paper is very well written.
I have only one remark: The use of chirp-z transform in improving frequency estimation is even reported in classical signal processing books such as the Proaky's one. I consider that your paper is very complete because of the consideration of termal phenomena. Could you please add a disclaimer on your contribution telling chirp-z transform has been already used in improving the frequency but remarking your differences with respect to previous works, such as your remarkable results?
Answer: Thanks very much for your approval of our paper.
We agree with you that the CZT has been used and reported to improve the frequency estimation performance and the submitted manuscript has also cited some of the research results. Nowadays, the most popular method is as following. Firstly, the spectrum with a higher frequency resolution is obtained by utilizing CZT, and then the frequency is estimated by searching the peak location. This processing may improve the frequency estimation performance to a certain extent, but on view of the fact that the real frequency of signal is continuous, and the estimated frequency must be the integral multiples of the spectrum resolution. As mentioned above, the methods based on the peak searching of CZT spectrum are obviously biased estimators, unless the real frequency exactly equals to the integral multiples of spectrum resolution. Different from the works of others, our proposed method is the innovation. Herein, we presented the theoretical expressions of frequency by deducing the mathematic relationship of three largest CZT spectrum samples and constructed an unbiased frequency estimator.
According to your constructive suggestion, we supplemented some research results, including the Proaky’s theory of Z-transform.
The original expression in the submitted manuscript is as following. “Besides the FFT spectrum, the Chirp-Z Transform (CZT) is also commonly used for frequency estimation [17,18]. While the current methods make the frequency estimation mainly by searching the peak line position of CZT spectrum, the accuracy of which is directly constrained by the CZT spectrum resolution, and more importantly, it is a biased estimation.”
The modified version is expressed as following.
Besides the FFT spectrum, the Chirp-Z Transform (CZT) is also commonly used for frequency estimation. Proakis (2021) [17] described the theory of CZT in detail. And Granados-Lieberman (2009) [18], Chen (2010, 2021) [19, 8], Zhang (2019) [6] have already introduced the CZT to improve the frequency estimation performance. The current methods estimate the frequency, which are still mostly dependent on searching the peak line position of CZT spectrum, meanwhile, making the estimation accuracy directly constrained by the CZT spectrum resolution. More importantly, the real frequency of signal is continuous, because of the fence effect of CZT, the estimated frequency must be the integral multiples of the CZT resolution. Therefore, the methods we mentioned above, which are based on the peak line position searching of CZT spectrum, are obviously biased estimators. Herein, we presented the theoretical expressions of frequency by deducing the mathematic relationship of three largest CZT spectrum samples and constructed an unbiased frequency estimator.
Question 2
Please note that there is a bold text in page 3 signaling an error between square brackets.
Answer: We highly appreciate you for reminding of the wrong bookmark in page 3. During the manuscript preparation, we used the cross-reference function of Microsoft Office Word when referring the literatures. Before submission, we modified the format according to the template and changed the citation index number to corresponding number. It is during the process when we omit the bookmark in page 3. Now we have already corrected the error.
